# Stroke Volume Variation and Stroke Volume Index Can Predict Fluid Responsiveness after Mini-Volume Challenge Test in Patients Undergoing Laparoscopic Cholecystectomy

**DOI:** 10.3390/medicina56010003

**Published:** 2019-12-19

**Authors:** Eun-Jin Moon, Seunghwan Lee, Jae-Woo Yi, Ju Hyun Kim, Bong-Jae Lee, Hyungseok Seo

**Affiliations:** 1Department of Anesthesiology and Pain Medicine, Kyung Hee University Hospital at Gangdong, College of Medicine, Kyung Hee University, Seoul 05278, Korea; loftylion@naver.com (E.-J.M.); mdyjwchk@khu.ac.kr (J.-W.Y.); tenten111@naver.com (J.H.K.); lbj8350@naver.com (B.-J.L.); 2Department of Surgery, Kyung Hee University Hospital at Gangdong, College of Medicine, Kyung Hee University, Seoul 05278, Korea; histones1@khnmc.or.kr

**Keywords:** fluid responsiveness, stroke volume variation, pneumoperitoneum, reverse Trendelenburg position, mini-volume challenge test

## Abstract

*Background and Objectives*: For using appropriate goal-directed fluid therapy during the surgical conditions of pneumoperitoneum in the reverse Trendelenburg position, we investigated the predictability of various hemodynamic parameters for fluid responsiveness by using a mini-volume challenge test. *Materials and Methods*: 42 adult patients scheduled for laparoscopic cholecystectomy were enrolled. After general anesthesia was induced, CO_2_ pneumoperitoneum was applied and the patient was placed in the reverse Trendelenburg position. The mini-volume challenge test was carried out with crystalloid 4 mL/kg over 10 min. Hemodynamic parameters, including stroke volume variation (SVV), cardiac index (CI), stroke volume index (SVI), mean arterial pressure (MAP), and heart rate (HR), were measured before and after the mini-volume challenge test. The positive fluid responsiveness was defined as an increase in stroke volume index ≥10% after the mini-volume challenge. For statistical analysis, a Shapiro–Wilk test was used to test the normality of the data. Continuous variables were compared using an unpaired *t*-test or the Mann–Whitney rank-sum test. Categorical data were compared using the chi-square test. A receiver operating characteristic curve analysis was used to assess the predictability of fluid responsiveness after the mini-volume challenge. *Results*: 31 patients were fluid responders. Compared with the MAP and HR, the SVV, CI, and SVI showed good predictability for fluid responsiveness after the mini-volume challenge test (area under the curve was 0.900, 0.833, and 0.909, respectively; all *p*-values were <0.0001). *Conclusions*: SVV and SVI effectively predicted fluid responsiveness after the mini-volume challenge test in patients placed under pneumoperitoneum and in the reverse Trendelenburg position.

## 1. Introduction

Laparoscopic surgery is widely performed because of its advantages with smaller surgical wounds, attenuated pain, and early recovery [1,2]. Currently, to improve surgical outcomes [3], to decrease postoperative pulmonary complications, and to enhance patient recovery [4], intraoperative fluid management is regarded as critical.

Goal-directed fluid therapy (GDFT) aims to maintain central euvolemia based on hemodynamic parameters. During GDT, stroke volume variation (SVV), which is a variation in stroke volume (SV) induced by cyclic intrathoracic pressure changes during mechanical ventilation [5], can be helpful in predicting fluid responsiveness, as reported in previous studies [6,7,8]. Laparoscopic cholecystectomy requires pneumoperitoneum and is performed in the reverse Trendelenburg position, which can alter the usefulness of the SVV in predicting fluid responsiveness via changes in intrathoracic pressure and systemic venous return. Furthermore, laparoscopic cholecystectomy is a hemodynamically stable surgery and requires just a small amount of fluid. Therefore, conventionally used fluid challenge protocol with 500 mL of colloid or 1000 mL of crystalloid would be inappropriate or could result in volume overload during laparoscopic cholecystectomy. In the present study, we used a mini-volume challenge test using 4 mL/kg of crystalloid and investigated the ability of SVV to predict fluid responsiveness under particular conditions of pneumoperitoneum in the reverse Trendelenburg position.

## 2. Methods

### 2.1. Patients

This study was approved by the Kyung Hee University Hospital at the Gangdong Institutional Review Board (IRB KHNMC201706020-HE001, approved date: 28/12/2017), and written informed consent was obtained from all individual participants included in the study. The trial was registered before patient enrollment at http://cris.nih.go.kr (KCT0002962). This study adhered to the applicable STROBE guidelines. A total of 42 adult patients scheduled for laparoscopic cholecystectomy were enrolled, and written informed consent was obtained from them. The exclusion criteria included body mass index (<18.5 kg/m^2^ or >35 kg/m^2^); patients with any gross deformity of the rib cage or spine; patients with heart disease (valvular heart disease, coronary heart disease, known arrhythmia, decreased ejection fraction, or suspicion of ischemic heart disease); patients showing serum creatinine (>1.4 mg/dL); and patients receiving hemodialysis or peritoneal dialysis.

### 2.2. Surgical Procedure

A pneumoperitoneum was created using the Veress needle technique, and a 5 mm trocar was placed in the umbilical region to facilitate the insertion of a 5 mm video scope. Intra-abdominal pressure (IAP) was set at 12 mmHg, and rate of insufflations was increased to 2 to 3 L/min. The patient was placed in a supine position with a 15–20 degrees reverse Trendelenburg tilt and 10 degrees lift on the right side. The epigastric 12 mm port was made, the third 5 mm port was inserted at the mid-clavicular line, and the fourth 5 mm port was cannulated at the right mid-axillary line at the level of the umbilicus.

### 2.3. Anaesthesia Protocol

General anesthesia and patient monitoring were performed according to our institutional standards. Intraoperative monitoring included electrocardiography, intra-arterial blood pressure, end-tidal carbon dioxide concentration, and peripheral oxygen saturation. Anesthesia was induced with propofol (2 mg/kg) and target-controlled infusions of remifentanil (Orchestra^®^ Base Primea; Fresenius Kabi, Bad Homburg, Germany) with effect-site concentrations of 3 ng/mL. Rocuronium bromide (0.6 mg/kg) was given to facilitate tracheal intubation. Anesthesia was maintained with 3–6% desflurane and 50% oxygen in the medical air. The effect-site concentrations of remifentanil target-controlled infusions were adjusted between 2 and 5 ng/mL. The depth of anesthesia was monitored using the bispectral index (A-1050 monitor; Aspect Medical Systems, Newton, MA, USA) and maintained between 40 and 60. Desflurane administrations and remifentanil target-controlled infusions were intermittently adjusted during surgeries according to the bispectral index or hemodynamic parameters but were not changed during fluid challenges or hemodynamic measurements. Volume-controlled mechanical ventilation using an anesthesia machine (Primus^®^; Dräger, Lübeck, Germany) was performed and set with a fixed tidal volume of 8 mL for the ideal body weight and a respiratory rate of 10–16 per minute to maintain an end-tidal carbon dioxide concentration between 35–40 mmHg. A positive end-expiratory pressure of 5 cmH_2_O was applied in all patients. During the main procedure, all patients were placed in the reverse Trendelenburg with their right side raised. Pneumoperitoneum was achieved by continuous carbon dioxide insufflation, maintaining a 12 cmH_2_O of IAP.

### 2.4. Hemodynamic Monitoring

After the induction of anesthesia, the radial artery was catheterized for continuous arterial pressure monitoring, and an additional intravenous route for the fluid challenge was secured with a 16-gauge catheter. An indwelling radial artery catheter was connected to the hemodynamic monitoring system (EV1000; Edward Lifesciences Corp., Irwin, CA, USA) via FloTrac™ (Edwards Lifesciences Corp.) sensors. Cardiac output and SV were calculated by real-time arterial waveform analysis. The SVV was computed over 20 s using the following formula:SVV (%) = 100 × (SV_max_ − SV_min_)/SV_mean_(1)

After zeroing the atmosphere, cardiac output, SV, and SVV values were obtained continually by arterial waveform analysis. SV_max_, SV_min_, and SV_mean_ indicate the maximal, minimal, and mean stoke volume in a respiratory cycle, respectively.

### 2.5. Fluid Challenge Protocol

The intraoperative basal fluid was administered at a rate of 2 mL/kg/h of crystalloid solution. Hemodynamic parameters were measured just before and 1 min after the fluid challenge. The fluid challenge was carried out with 4 mL/kg of crystalloid solution (plasma solution A; CJ, Seoul, Korea) over 10 min via an infusion pump. During the fluid challenge, the respiratory rate and anesthetic concentration was not changed. 

### 2.6. Outcome Measurement

Patients who achieved a stroke volume index (SVI) increase ≥10% after fluid challenge were classified as fluid responders and the others as fluid nonresponders. At each timepoint, the cardiac index (CI), mean arterial pressure (MAP), heart rate (HR), SVI, and SVV were recorded. Our primary outcome was the predictability of each hemodynamic parameter for fluid responsiveness after the mini-volume challenge. For the secondary outcome, we investigated the hemodynamic changes before and after the mini-volume challenge.

### 2.7. Statistical Analysis

The sample size was determined using the difference between the area under the curve at 0.75 (the alternative hypothesis that the SVV can predict fluid responsiveness) and 0.5 (null hypothesis). Assuming a type I error of 0.05 and the desired power of 0.80, 38 patients were required for the analysis. Expecting a dropout rate of 10%, we enrolled 42 patients. A Shapiro–Wilk test was used to test the normality of the data. Continuous variables were compared using an unpaired *t*-test or the Mann–Whitney rank-sum test. Categorical data between the two groups were compared using the chi-square test or Fisher’s exact test. A paired *t*-test was used to compare the changes before and after the fluid challenge. In order to compare the changes in hemodynamic variables simultaneously at each timepoint and the differences between responders and nonresponders, the two-way repeated measure analysis of variance was used. The receiver operating characteristic curve (ROC) analysis was performed to assess the predictability of fluid responsiveness after the fluid challenge for each hemodynamic variable (SVV, MAP, SVI, and HR). The optimal cut-off value was determined using a value based on the Youden index, which was calculated at the maximum (sensitivity + specificity -1). All results were expressed as mean ± standard deviations, medians (interquartile range), or numbers (%). A *p*-value <0.05 was regarded as statistically significant. Statistical analysis was performed using MedCalc^®^ version 13.2.0 (MedCalc Software; Ostend, Belgium) or SigmaPlot 10.0 (Systat Software, Inc.; San Jose, CA, USA).

## 3. Results

Among 42 patients, 31 patients were fluid responders (73.8%). Demographics and perioperative data of patients are shown in Table 1. Between nonresponders’ and responders’ age, sex, weight, and height, the preoperative laboratory test did not show any statistical differences. Additionally, perioperative data including anesthesia time, postoperative nausea, vomiting, and administered fluid amount showed no significant differences in variables between fluid responders and fluid nonresponders.

The serial changes in hemodynamic parameters during surgery are shown in Figure 1. Particularly, changes before and after the mini-volume challenge are shown in Table 2. Before and after mini-volume challenge, SVV changed from 8 ± 4 to 6 ± 4, CI from 3.4 ± 0.7 to 3.0 ± 1.0, and SVI from 44 ± 9 to 43 ± 3 in nonresponders. However, in responders, SVV changed from 16 ± 6 to 6 ± 4, CI from 2.5 ± 0.6 to 3.3 ± 0.8, and SVI from 28 ± 7 to 41 ± 10 after mini-volume challenge test (*p* = 0.0001, 0.0012, and 0.0001, respectively).

The ROC curve of each parameter is shown in Figure 2. The area under the curve of each hemodynamic parameter for the prediction of fluid responsiveness after the mini-volume challenge is provided in Table 3. The area under the curve of the SVV, CI, and SVI for predicting fluid responders was 0.900, 0.833, and 0.909 (*p* <0.0001). The optimal cut-off values for predicting an SVI increase of ≥10% after the mini-volume challenge in each parameter was ≥11% for the SVV (sensitivity 80.7%, specificity 90.8%); ≤2.7 mL/min/m^2^ for the CI (sensitivity 70.0%, specificity 81.8%); and ≤40 mL/m^2^ for the SVI (sensitivity 96.8%, specificity 72.7%).

## 4. Discussion

In the present study, the SVV, as well as the SVI and CI, showed good predictability for fluid responsiveness after the mini-volume challenge test in patients undergoing laparoscopic cholecystectomy under pneumoperitoneum and being placed in the reverse Trendelenburg position.

The GDFT is composed of minimal maintenance fluid therapy and additional volume administrations guided by fluid responsiveness [3]. Excessive fluid administration can cause hypervolemia, glycocalyx damage, and increased vascular permeability [9]. In contrast, fluid restriction can aggravate hypovolemia and tissue hypoperfusion [10]. Thus, the balance between volume deficit and overload is critical in fluid management. In previous studies, GDFT has contributed to the reduction of postoperative complications and mortalities by monitoring cardiac output for maintenance of tissue perfusions [11,12,13].

The SVV has been proven to be a reliable dynamic indicator for predicting fluid responsiveness without the implementation of the fluid challenge [7,8,14]. Theoretically, positive pressure ventilation induces cyclic changes in intrathoracic pressure, thereby changing SV, and this variability in SV can be used as the dynamic volume index in GDFT. During pneumoperitoneum, increased IAP and reduced chest wall compliance compress cardiovascular structures, leading to changes in pleural pressure variations and interruptions of venous return to the heart, resulting in SV decrease regardless of changes in blood volume. Moreover, particularly in laparoscopic cholecystectomy, the surgical position of reverse Trendelenburg may provide further reduction in the venous return and aggravate SV-decrease through reduced preload. Therefore, under pneumoperitoneum and in the reverse Trendelenburg position, conventionally used parameters for GDT should be adjusted.

In the present study, a relatively small volume of crystalloid (4 mL/kg) was administered to perform fluid challenge. Conventionally, the fluid challenge technique for investigating fluid responsiveness uses 500 mL of colloid [15] or 7–10 mL/kg of crystalloid [16]. Although the operational conditions of laparoscopic cholecystectomy includes pneumoperitoneum and the reverse Trendelenburg position and both can cause an excessive preload-dependent state, considering laparoscopic cholecystectomy is a less-invasive surgery with minimal bleeding risks, the volume for conventional fluid responsiveness test can be relatively large and may cause a volume overload postoperatively. Although the mini-volume test uses relatively small volumes, it has been used in many previous studies and showed good results [17,18,19].

Our results showed that the SVV, CI, and SVI showed good predictability of the SVI increase (≥10% after mini-volume challenge). Considering CI is derived from the SVI and HR, both the CI and SVI are closely associated with preload-dependent states anticipated during pneumoperitoneum and in the reverse Trendelenburg position. The SVV, which has the diagnostic threshold for fluid responsiveness between 11% and 13%, with a sensitivity of 0.82 and specificity of 0.86 [5], also showed good predictability for fluid responsiveness with high sensitivity and specificity. In our data, even using the mini-volume challenge test, the SVV and SVI can effectively predict fluid responsiveness in laparoscopic cholecystectomy under pneumoperitoneum and in the reverse Trendelenburg position. The cut-off value of the SVV was 11% in our results and was similar to previous studies. Its similarity means that the effect of increased IAP and the reverse Trendelenburg position in the fluid responsiveness index may be minimal in the present study. Generally, IAP can be maintained at 15 mmHg during laparoscopic cholecystectomy; however, we used 12 mmHg of IAP in the present study, and it may not be sufficiently high enough to reduce venous return. Moreover, despite that the SVI can be reduced by pneumoperitoneum and the reverse Trendelenburg position [20,21], another study reported that hemodynamic depression was mainly caused by general anesthesia. Therefore, the effects of increased IAP, even in addition to the reverse Trendelenburg position, caused minimal hemodynamic effects [22].

The present study had several limitations. First, in the present study, we used an uncalibrated system for the SV measurement technique. The uncalibrated SV measurement using the pulse contour analysis of the arterial waveform could be limited in hemodynamically stable patients, thereby being less valuable than using echocardiography or pulmonary arterial catheters. Although pulmonary arterial catheterization or transesophageal echocardiography can provide more reliable hemodynamic parameters, it is clinically too invasive and requires a high medical cost to use in daily practice or anticipated hemodynamically stable surgery. Second, the SVV measurements using the pulse contour analysis technique may be inaccurate in patients with arrhythmia from abnormal chest or abdomen diseases. Thus, it should be a clinical concern when interpreting the SVV for fluid management. Third, we enrolled patients undergoing laparoscopic cholecystectomy. Since laparoscopic cholecystectomy requires a relatively short procedure time and is a low-risk surgery with less hemodynamic changes, a zero-fluid balance strategy would be appropriate, and GDT may not be necessary. Even invasive arterial monitoring with FloTrac™ sensors is not required during laparoscopic cholecystectomy, in contrast with other major surgeries, such as esophagectomy or hepatectomy. However, it requires both pneumoperitoneum and the reverse Trendelenburg position. The results from these conditions would be helpful to apply in other major surgeries requiring GDT. Fourth, although SVV and SVI were useful for predicting fluid responsiveness after the mini-volume challenge test in laparoscopic cholecystectomy during our study, the small sample size could limit its clinical implications, and physicians should be careful when applying the present results to other clinical situations.

## 5. Conclusions

In conclusion, by using a mini-volume challenge test, the SVV, CI, and SVI can be useful predictors of fluid responsiveness in patients placed under pneumoperitoneum and in the reverse Trendelenburg position.

## Figures and Tables

**Figure 1 medicina-56-00003-f001:**
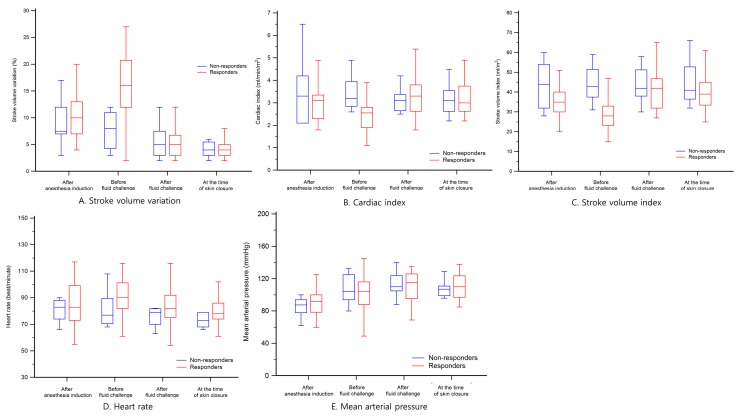
Changes in hemodynamic parameters during surgery. Each parameter was sequentially measured at four times: after anesthesia induction, before fluid challenge, after fluid challenge, and at the time of skin closure. (**A**) stroke volume variation; (**B)** cardiac index; (**C**) stroke volume index; (**D**) heart rate; and (**E**) mean arterial pressure.

**Figure 2 medicina-56-00003-f002:**
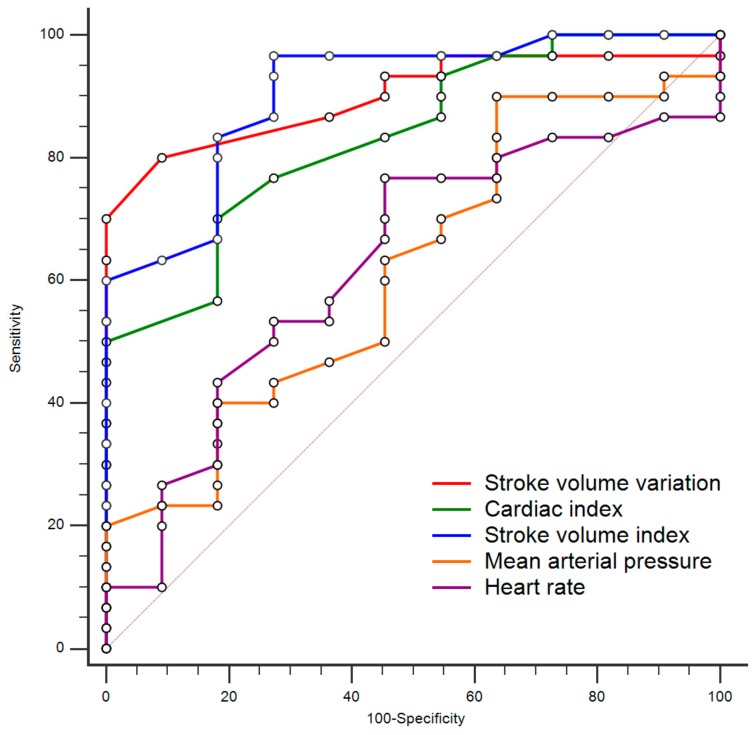
Comparisons between the receiver operating characteristic curves of each parameter.

**Table 1 medicina-56-00003-t001:** Demographic and perioperative data.

Variables	Nonresponders (*n =* 11)	Responders (*n =* 31)
Age (years)	55 ± 17	54 ± 11
Sex (male)	6 (54.5%)	18 (58.1%)
Weight (kg)	64.5 ± 12.7	67.7 ± 14.2
Height (cm)	160.1 ± 8.9	162.0 ± 10.3
Serum albumin (g/dL)	3.9 ± 0.6	4.2 ± 0.4
Serum creatinine (mg/dL)	0.69 ± 0.09	0.79 ± 0.18
Serum hemoglobin (g/dL)	13.2 ± 1.1	14.1 ± 1.8
Perioperative data		
Anaesthesia time (min)	60 (52–65)	55 (50–60)
Fluid, total administered (mL)	350 (300–432)	330 (319–387)
Gas out time (hours after surgery)	30.2 ± 21.8	30.4 ± 17.9
Postoperative nausea incidence		
at PACU/ until postoperative 24 h	0 (0%)/4 (36.4%)	1 (3.2%)/9 (29.0%)
Postoperative vomiting incidence		
at PACU/ until postoperative 24 h	0 (0%)/1 (9.1%)	0 (0%)/2 (6.4%)

Data expressed as mean ±SDs, medians (interquartile range), or numbers (%). PACU = postoperative care unit.

**Table 2 medicina-56-00003-t002:** Descriptive statistics of hemodynamic changes before and after mini-volume challenge.

Variables	Nonresponders (*n =* 11)	Responders (*n =* 31)
Stroke volume variation (%)		
before fluid challenge	8 ± 4	16 ± 6 *
after fluid challenge	6 ± 4	6 ± 4 ^†^
Cardiac index (L/m^2^)		
before fluid challenge	3.4 ± 0.7	2.5 ± 0.6 *
after fluid challenge	3.0 ± 1.0	3.3 ± 0.8 †
Stroke volume index (mL/m^2^/beat)		
before fluid challenge	44 ± 9	28 ± 7 *
after fluid challenge	43 ± 3	41 ± 10 †
Heart rate (beat/min)		
before fluid challenge	82 ± 13	90 ± 17
after fluid challenge	77 ± 3	82 ± 14 †
Mean arterial pressure (mmHg)		
before fluid challenge	108 ± 18	100 ± 24
after fluid challenge	112 ± 16	111 ± 18 †

Data expressed as medians (interquartile range), with * = *p* <0.05, compared with nonresponders, and † = *p* <0.05, compared with after the fluid challenge.

**Table 3 medicina-56-00003-t003:** Hemodynamic parameters’ predictability of fluid responsiveness after the mini-volume challenge test.

Variables	Area Under Curve (95% CI)	*p*	Cut-off Value	Sensitivity	Specificity
Stroke volume variation (%)	0.900 (0.766–0.971)	<0.0001	11	80.7%	90.9%
Cardiac index (mL/min/m^2^)	0.833 (0.684–0.931)	<0.0001	2.7	70.0%	81.8%
Stroke volume index (ml/m^2^)	0.909 (0.779–0.976)	<0.0001	40	96.8%	72.7%
Mean arterial pressure (mmHg)	0.617 (0.452–0.764)	0.254	123	90.3%	36.4%
Heart rate (beats/min)	0.639 (0.462–0.773)	0.134	77	77.4%	54.6%

CI = confidence interval, and sensitivity or specificity is derived from the cut-off value.

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
