# Peer review of "Stroke Volume Variation and Stroke Volume Index Can Predict Fluid Responsiveness after Mini-Volume Challenge Test in Patients Undergoing Laparoscopic Cholecystectomy"

_medicina, 2019, doi:10.3390/medicina56010003_

Round 1
Reviewer 1 Report
Hemodynamic parameters predicting fluid responsiveness after mini-volume challenge test in patients undergoing laparoscopic surgery in reverse Trendelenburg position
This study has investigated the predictability of various hemodynamic parameters for fluid responsiveness by using a mini-volume challenge test. The mini-volume challenge test with crystalloid 4 ml/kg over 10 minutes was conducted in forty-two adult patients scheduled for laparoscopic cholecystectomy. Thirty-one patients showed an increase in stroke volume index (SVI) ≥10% after the mini-volume challenge and were considered as fluid responders. Moreover, stroke volume variation (SVV) and SVI were found to be better predictors of fluid responsiveness in the min-volume challenge test as compared to other hemodynamic parameters. Overall, this is a well-planned study. The results are clear and easy to understand. The title is very wordy and vague. It does not entirely reflect the conclusion of the study.
Author Response
Thank you for your kind comment. Please see the attachment.

Reviewer 2 Report
The authors describes about hemodynamic parameters during laparoscopic surgery, especially in reverse Trendelenburg position. However, this study have demonstrate the only baseline parameters under pneumoperitoneum. Anesthesiologists and Surgeons needs the true parameters such as changes during bleeding and compression effects by pneumoperitoneum. If possible, is there any data to be revealed as more clinically useful information. And, both A-line and FloTrac are not routinely used in general laparoscopic surgery. These device are used in esophagectomy, major hepatectomy, PD, and necessity due to patient's condition. Authors have to describe about these discrepancy more strictly.
Author Response

(The authors gave the same response as above.)

Reviewer 3 Report
Dear Authors and Editorial Team-
It is very interesting to see the predictability of various hemodynamic parameters for fluid responsiveness by using mini-volume challenge test.
Here are some comments need your attentions.
Abstract- please add brief info about analysis. Eg A Shapiro-Wilk test, t-test or Mann–Whitney rank-sum test, chi-square test or Fisher’s exact test etc
Methods- Outcome Measurement- mention primary and secondary outcomes in details and definitions of them. Use small italic p and not capital P Results: Mention demographic characteristics, Descriptive statistics of hemodynamic changes before and after mini-fluid challenge of Non-responders and responders in brief in a paragraph Results: The ROC curve of each parameter … In this paragraph, pls add info from table 3 regrading sensitivity and specificity to explain them to readers. Discussion: comment on small sample size and impact of it on outcomes in limitation. Thank You. Regards UAuthor Response
Thank you for your kind comment. Please see the attachment.

Round 2
Reviewer 2 Report
Required data and sentences have been added in the revised manuscript. I think it is worth to be published.